**EMBO** *reports*

# Molecular mechanisms of co-infections

Philipp Darius Konstantin Walch [ID] ✉ & Petr Broz [ID] ✉

## Abstract

Co-infections generally cause exacerbated pathologies in patients, yet a knowledge gap between clinical data and the underlying molecular mechanisms remains. Clinical studies focus on patient outcome, but much less is known about molecular mechanisms and convergence points that define the interaction between different pathogens. In this review, we will summarize the current standing of the literature at the various scales of magnitude that co-infections impact: epidemiology, clinical observations, tissue- and organ-specificity, the single-cell level, and molecular mechanisms. Given the scarcity of systematic research across systems, we will focus on molecular interaction points that have been identified, comment on their generalizability and, where required, extrapolate from single-pathogen studies. More research of the host–pathogen–pathogen interface is direly warranted, and we hope to inspire advances addressing the intricate network between two co-occurring pathogens and their host. In addition to direct implications for co-infections, acquiring a better understanding of how microorganisms interact in this complex environment will enable us to better understand single-pathogen infections as well, which can lead to the development of novel treatment approaches.

**Keywords** Co-infection; Host–pathogen Interactions; Immune Response; Systems Biology
**Subject Categories** Immunology; Microbiology, Virology & Host Pathogen Interaction; Signal Transduction

## Introduction: the vast dimensionality and complexity of co-infections

The interface between pathogens and the host spans all scales of magnitude: it ranges from molecular interactions between pathogen effector proteins and their host targets all the way to clinical studies and entire ecosystems. In-between lie cellular effects that determine the spread or clearance of an infection, paracrine and systemic disruptions that involve a broad spectrum of host cell and tissue types, as well as a multitude of microbes (Fig. 1). This is especially true for co-infections, where multiple different pathogenic microbes infect the same host organism or even the same host cell at the same time (Pasman, 2012), often leading to exacerbated disease outcome, as well as superinfections, which are characterized by a temporal gap between the primary and the secondary infection

(Nowak and May, 1994). The interactions between two or more pathogens and their host are, as we will discuss, highly dimensional, and studies at all different scales have tried to assess and dissect this interconnectivity.

At the same time, it is essential to better understand how pathogens interact with one another and their host, and how infection routes can be disrupted using novel treatment strategies (Hancock and Sahl, 2013). Furthermore, due to the immense degree of co-evolution that pathogens have undergone, we can harness their pathogenicity mechanisms to learn more about host cell biology (Le Pendu et al, 2014; Kelly et al, 2024; Woolhouse et al, 2002). While there has been extensive research in single-pathogen studies, elucidating infection mechanisms, host–pathogen interactions and strategies to disrupt infection routes, comparably little knowledge has been acquired for co-infections (Lian et al, 2022). Furthermore, the single-pathogen infection is a rarely occurring case, and more frequently used as a simplified model to study infections in vitro. Therefore, we need to study this host–pathogen-pathogen interface in an interdisciplinary and methodologically diverse manner (Devi et al, 2021). In this review, we will provide an overview of the knowledge that has been generated with respect to co-infections, focusing mainly on the molecular and cellular interaction points that impact the infection outcome. We will highlight and discuss various model systems that are being used to dissect the host–pathogen interface during (co-)infections, and will try to conclude the importance of this work, not only to the scientific field, but also beyond.

## Level 1: systems and organisms: epidemiology, occurrence and clinical impact of co-infections

The dramatic impact that co-infections can have on patients has been broadly described using clinical and observational studies: In the majority of reported cases, co-infections led to a more severe disease outcome, an exacerbation of symptoms, higher morbidity and mortality, especially in vulnerable groups, such as children, the elderly and the immunocompromised (Qiao, 2023; Zhou et al, 2023; Potgieter et al, 2023). It is therefore possible that the spectrum of clinical studies underestimates the occurrence of antagonistic interactions during co-infection, since milder symptoms might not be reported and existing research is hence blind to these cases (Box 1). Co-infection can be categorized by tissue or organ that they affect: pathogens that affect the immune system directly (Palefsky and Holly, 2003; Martin-Loeches et al, 2017; Bell and Noursadeghi, 2018), respiratory infections (Meskill and O'Bryant, 2020), as well as gastrointestinal (GI) pathologies (Lian et al, 2022) are most common (Fig. 2). More rarely, co-infections that occur in the liver (Blackard and Sherman, 2008; Chou et al,

Department of Immunobiology, University of Lausanne, Chemin des Boveresses 155, CH-1066 Epalinges, Switzerland. ✉E-mail: philipp.walch@unil.ch; petr.broz@unil.ch

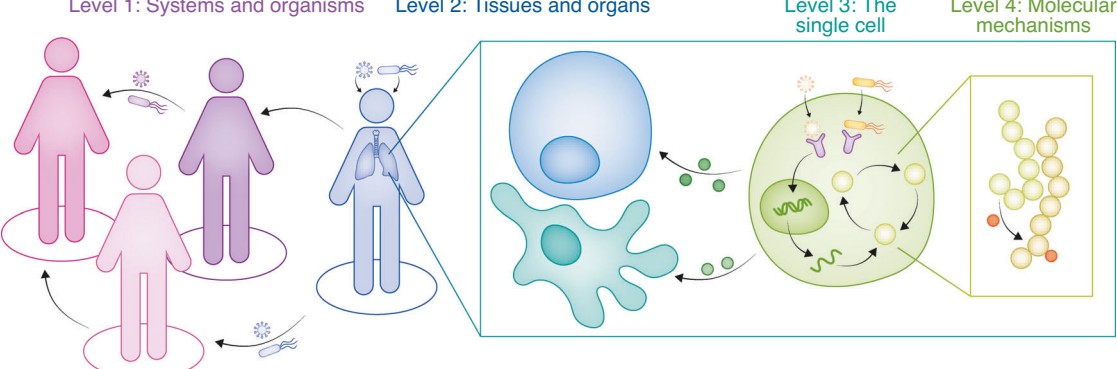

**Figure 1. Co-infections across scales.**

Ranging from molecular mechanisms to entire ecosystems, co-occurring pathogens impact each other, as well as the host. Assessing the impact of co-infections on various levels with unbiased Systems Biology approaches helps dissect the intricate network of host–pathogen-pathogen interactions. Level 1: co-infections alter the clinical outcome for patients, as well as the epidemiological properties of an infectious disease. Level 2: on a tissue- or organ-specific level, co-occurring pathogens interact with each other and reshape the host through tissue disruption, modification of the cytokine landscape or nutrient microenvironment. Level 3: pathogens can simultaneously infect the same cell and alter detection of co-occurring microbes, transcription, translation or cell signaling. Level 4: by modifying how proteins interact with each other, how signals are processed and how cell defenses are regulated, pathogens impact each other on a molecular level during co-infection.

**Box 1    In need of answers**

- What is the main scale of pathogen-interaction during co-infection: systemic, on a tissue level or within a single host cell?

- Is there a blind spot for antagonistic interactions (i.e., improved disease outcome), since these cases might be under-reported

- Is there an evolutionary benefit for pathogens to induce co-infections with other pathogens? If so, what are these?

- What are the best treatments options for co-infections and can a better understanding of the molecular basis of co-infection help to identify new therapeutical approaches?

2022), the brain (Garg et al, 2012), or in the bladder (Cheung et al, 2020) have been described.

Due to its dramatic effect on the immune system, Human Immunodeficiency Virus (HIV) infection, when untreated, leading to Acquired Immunodeficiency Symptom (AIDS), has been thoroughly studied with respect to its pathology in co-infections (Rockstroh and Spengler, 2004; Sharan et al, 2020; Lloyd-Smith et al, 2008; Naranbhai et al, 2014). The mechanism by which HIV exerts its devastating impact on co-infections is due to its specific targeting of CD4-positive T-cells upon infection, leading to their depletion (Deeks et al, 2015; Rosenberg and Fauci, 1989). The resulting demise of the immune response predisposes untreated HIV-positive patients for secondary infection with many different pathogens, ranging from viruses (e.g., Hepatitis B and C), bacteria (e.g., *Mycobacterium tuberculosis*) and fungi (e.g., *Candida albicans*, *Cryptococcus neoformans*) to helminths and other parasites (e.g., *Plasmodium falciparum*) (Chang et al, 2013) (Fig. 2). These secondary/co-infections can massively exacerbate symptom severity, morbidity and mortality in patients (Palefsky and Holly, 2003). This is especially true for pathogens that are cleared

by CD4-positive T-cells in HIV-negative patients, such as *Mycobacterium tuberculosis* (Bell and Noursadeghi, 2018). In addition, *Plasmodium falciparum*, the causing agent of malaria has been shown to promote secondary bacterial infections, predominantly with *Salmonella* (Wilairatana et al, 2022), as well as parasitic and helminth infections (Wudneh et al, 2021; Ornellas-Garcia et al, 2023).

In the GI setting, Rotavirus and Norovirus are clinically relevant infectious agents, for which secondary infections with other enteric pathogens have been described: *Escherichia coli* co-infection has been associated with exacerbated gastrointestinal symptoms in children (Grimprel et al, 2008; Bhavnani et al, 2012), and *Clostridioides difficile* displayed an increased bacterial burden (Valentini et al, 2013; Stokely et al, 2016) upon viral–bacterial co-infection (Fig. 2). Only a few studies have investigated the role of bacterial-bacterial co-infections, and the review of available clinical studies highlights key difficulties in determining the precise impact, such as small sample sizes and a discrepancy in the timing of diagnostic tests (de Graaf et al, 2015). It is however likely that dysbiosis introduced by GI pathogens can enhance the occurrence of opportunistic infections (Tewari and Dey, 2024). Furthermore, bacterial-helminth co-infection have been shown to increase parasite egg production and shedding (Lass et al, 2013), and display clinical relevance in *Helicobacter pylori* pathogenicity (Seid et al, 2018). Overall, studies in the intestinal setting have pointed to a diverse set of possible outcomes, ranging from mutual exclusivity, especially after viral infection, to increased susceptibility upon bacterial primary infection (Lian et al, 2022).

The occurrence of co-infections in the context of respiratory disease had for the longest time focused on Influenza-A Virus (IAV). Disease severity, as well as morbidity of IAV are dramatically increased by staphylococcal (Kishida et al, 2004; Cheng et al, 2009; Jia et al, 2018) or streptococcal (Martin-Loeches et al, 2017) co-infections (Melvin and Bomberger, 2016), and these pathogen pairs have also been investigated on a molecular scale, as

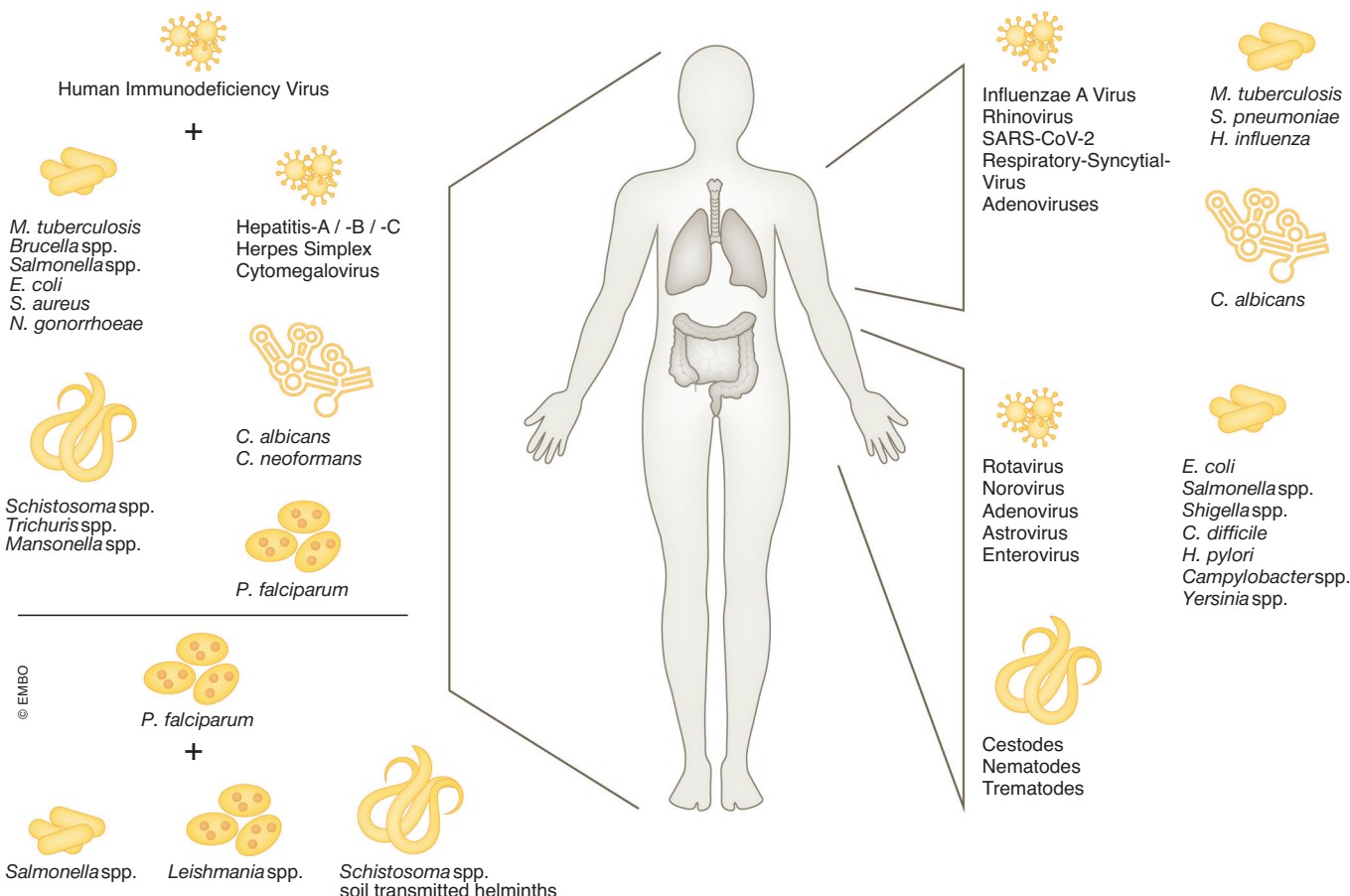

**Figure 2. Clinically relevant pathogen combinations during co- and superinfections.**

Various clinical studies have assessed the co-occurrence of pathogens in hospitalized patients. In most cases, co- and superinfections caused exacerbated symptoms and altered infection kinetics. Systemic infections with the Human Immunodeficiency Virus or *Plasmodium falciparum* lead to increased susceptibility to secondary infections (Brown et al, 2006; Boulougoura and Sereti, 2016; Khademi et al, 2018). On an organ-specific level, the lung and the gastrointestinal system are most relevant for co-infections: as barrier organs at the interface between our body and the environment, harboring a rich microbiome, including opportunistic pathogens (Brown et al, 2006; Varyani et al, 2017). Further references of listed pathogen pairs are mentioned throughout the text. Relevant abbreviations: spp: species pluralis, SARS-CoV-2 severe acute respiratory syndrome coronavirus 2.

discussed later. However, since the SARS-CoV-2 pandemic, more and more studies have interrogated what impact SARS-CoV-2 infection has on secondary pathogens (Langford et al, 2020; Zhu et al, 2020). Interestingly, in the case of SARS-CoV-2, these impacts can outlast the primary viral infection beyond its clearance and its primarily affected organ, the lung: reshaping host innate immunity has serious implications for a patient's response to secondary infections and opportunistic pathogens, such as *C. albicans,* even after the virus has been fully cleared (Moser et al, 2021; Alfaifi et al, 2024) (Fig. 2). Similar mechanisms of persevering reshaping of the host have been described for other respiratory pathogens, such as Influenza Virus (Didierlaurent et al, 2008).

Co-infections not only impact individuals but also have more global implications on populations as a whole, thereby exerting implications for epidemiology and transmissibility of infectious diseases. The geographic co-occurrence of pathogens, which can favor co-infection, has been described for, among others, pathogens causing severe respiratory disease, including Rhinovirus,

*Streptococcus pneumoniae* and *Haemophilus influenzae* (Jacoby et al, 2007), Enterotoxigenic *Escherichia coli* with Enteropathogenic *Escherichia coli* (EPEC) or *Campylobacter* (Colgate et al, 2023), and viral enteric pathogens, such as Rotavirus, Norovirus, Astrovirus, Adenovirus, and Enteroviruses (Makimaa et al, 2020) (Fig. 2). In addition, conditional co-infection, where a secondary pathogen requires a primary infection has been studied in the cases of Hepatitis Delta Virus, depending on Hepatitis B (Rizzetto et al, 1977; Negro and Lok, 2023), or plant pathogens which facilitate or require co-infection with a second microbe (Abdullah et al, 2017). These results indicate that co-occurrence, and potentially co-evolution of pathogens influence geographic spread and dissemination of pathogens.

Furthermore, there have been several epidemiological studies that underscore the relevance of pathogen co-occurrence on the infection dynamics within given populations (Susi et al, 2015; Kirschner, 1999), providing strong indications that co-infections lead to an increased spreading. Contributing factors to these altered

dynamics range from the facilitated spread during co-infection (Lass et al, 2013) to differences in replication efficiency during viral–viral co-infection (Williams et al, 2016) and highlight the complexity of pathogen–pathogen interactions. A final, global impact that the occurrence of co-infections has on a population, is the emergence and dissemination of antimicrobial-resistant (AMR) bacterial pathogens. Epidemiological studies on SARS-CoV-2 revealed a high occurrence of secondary AMR bacterial infections, especially with *Staphylococcus*, *Acinetobacter* or *Klebsiella* (Sathya-kamala et al, 2022; Marua et al, 2022). In these cases, as well as similar studies, the treatment with excessive amounts of antibiotics, due to the severe risk of exacerbated disease during SARS-CoV-2-bacterial co-infection has been attributed to this effect (Bengoechea and Bamford, 2020), and it is plausible that this holds true for other pathogens which display an increased incidence in co-infection with AMR bacteria.

Several of the effects that have been observed on the ecosystem level can be traced back to the impact that co-infections have on individual patients, as well as molecular dependencies of the two pathogens on each other. The scales of magnitude that we review here are therefore to be understood in the context of interconnectivity, rather than isolation. Co-infection with a given pair of pathogens has implications on several or even all levels, and more research is required to determine the importance and contribution of each of them (Box 1).

## Level 2: organs and tissues

While the infection with a single pathogen can cause serious damage to affected organs, such as the lung or the GI tract, the combined effect of a co-infection is in many cases even more deleterious. This has been evident in clinical studies, yet we are only beginning to investigate effects on a tissue- or organ-level, and the underlying role of cell-to-cell communication. The mechanistic understanding of disease outcome during co-infection is coming into focus, and especially for co-infections that occur in a tissue-specific manner, effects at the level of cell-to-cell communication and association are actively being investigated: for IAV co-infections with staphylococci and streptococci for example, the disruption of the epithelial barrier by the virus has been identified as a contributing factor to increased disease severity (Ruan et al, 2020, 2022). This is mainly due to the destruction of the protective layer which exposes more vulnerable tissues underneath and facilitates the colonization through bacterial pathogens by altering the mucus layer (Yang et al, 2014). In addition, modulation of the innate immune response has been described for IAV infections (Didierlaurent et al, 2008; Rynda-Apple et al, 2015).

Furthermore, cell-to-cell signaling, which is an essential component of healthy tissue physiology, and which can be significantly altered during (co-)infection impacts the severity of disease states. By upregulating specific markers to be expressed at the cell surface (Moon et al, 2024), as well as by inhibiting cell death (Lamkanfi and Dixit, 2010), pathogens protect their host cell niche. The altered behavior of the infected host cell then impacts neighboring (bystander) cells through direct cell–cell communication, as well as through the secretion of cytokines. The latter has been prominently described for the BCG-vaccine, which, in addition to conferring protection to tuberculosis, also modifies the responses of innate immune cells, leading to an increased protection toward a variety of secondary infections (Chen et al,

2023), including SARS-CoV-2, influenza or yellow fever. This mechanism has been termed "trained immunity" (Chen et al, 2023), and is actively being investigated and further refined for novel preventive and treatment strategies, including anti-cancer therapies (Jiang and Redelman-Sidi, 2022; Liatsos et al, 2025).

Lastly, pathogens actively modulate the extracellular niche of colonized organs and tissues: most prominently, *Salmonella* triggers the production of oxygen, nitrate, tetrathionate, and lactate through triggering and maintaining inflammation (Rogers et al, 2021), thereby gaining a competitive advantage over other enteric microbes. This further drives the inflammation of the intestine (Galán, 2021), attracts antigen-presenting cells, which serve as a niche (Niedergang et al, 2000), and causes dysbiosis by depleting the healthy microbiome (Gillis et al, 2018). The latter has severe ramifications for the blooming of opportunistic pathogens, such as *Clostridioides difficile* (Antharam et al, 2013), as well as the ability of other enteric pathogens to colonize the otherwise well-protected niche (Caballero-Flores et al, 2023). Furthermore, viral pathogens, such as IAV, can change the tissue microenvironment, causing redox imbalance, as well as an altered interferon response by macrophages and cells of the adaptive immune system (Sender et al, 2021).

## Level 3: the single cell

As presented in the previous paragraph, modulation of the host cell niche has severe impacts on neighboring host cells, yet it also impacts individual cells where it can be a deciding factor whether a cell is infected by two pathogens simultaneously: by inducing the surface expression of CD47, IAV protects the host from extrinsic killing, thereby protecting its niche. However, in a remarkable instance of co-evolution, the increased levels of accessible CD47 also allow for a more efficient attachment of *Staphylococcus aureus* to the virally infected cell (Moon et al, 2024), and hence exacerbate the impact of the bacterial secondary infection. Similar effects, the upregulation of a surface receptor by one pathogen, thereby promoting the attachment of a secondary infectious agent, have been described for other pathogens (Rossi et al, 2020), including RSV or *H. influenzae*, both upregulating ICAM-1 (Sajjan et al, 2006; Rossi and Colin, 2017).

While interactions like this indicate an increased permissiveness of an individual cell for a secondary infection upon pre-infection, there is no general directionality that can be concluded from the literature (Box 1). Other cellular factors, such as resource limitation or sequestration, have been described in the case of *Wolbachia* and Dengue Virus (Hoffmann et al, 2011; Caragata et al, 2014; Koh et al, 2020), leading to a mutual exclusivity of the two microbes (Fox et al, 2023). For virus–virus co-infections, mutual exclusivity has been described (e.g., Plum Pox Virus, Tobacco Vein Mottling Virus, and Clover Yellow Vein Virus in the plant system) (Syller, 2012) or Rhinovirus-SARS-CoV-2 co-infections, due to increased interferon response by the host (Dee et al, 2021), as well as co-dependence or conditional co-infection (e.g., Hepatitis B and -D) (Rizzetto et al, 1977; Negro and Lok, 2023). Also, for viral–bacterial co-infection, various dynamics and outcomes are observed, revealing either strong antagonisms (e.g., Norovirus and *Salmonella*) (Agnihothram et al, 2015) or synergies (e.g., Rice Yellow Mottle Virus and *Xanthomonas oryzae*) (Abdullah et al, 2017).

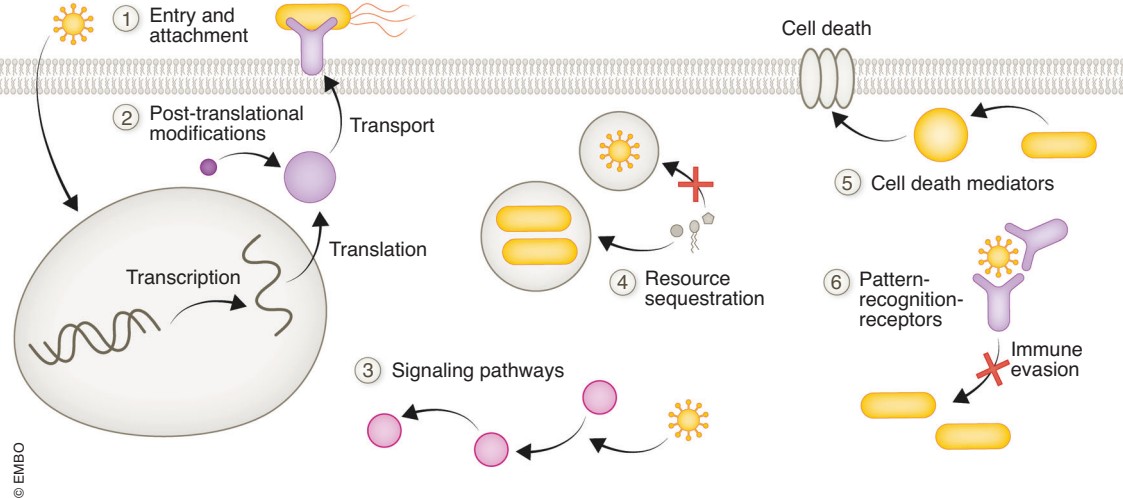

**Figure 3. Molecular interaction points during co-infection and general motivation for a Systems Biology approach in co-infections.**

Schematic of putative cellular interaction points for co-occurring pathogens in a host cell: attachment to the cell and invasion (1), in addition to "housekeeping" processes, such as transcription and translation (2), and host cell signaling (3) are commonly affected in single-pathogen infections, and have downstream effects on co-occurring pathogens. Furthermore, resource limitation or sequestration (4), the induction of cell death (5) or immune evasion (6) can mediate the interaction of two pathogens. While several of these interaction points have been studied in the setting of two co-occurring pathogens, the majority of the knowledge we have with respect to the role of innate immune signaling is based on research with single pathogens and subsequent determination of host responses.

In a recent study that attempts to assess the cellular effects during co-infection, an in vitro screening approach was employed to deduce general trends: overall, antagonistic interactions in cell death, as well as pathogen growth were more prominent (Walch and Broz, 2024), yet the translatability of these results into more complex systems remains to be shown by further studies. However, it is of note that the induction of cell death and inflammasome signaling has been highlighted and characterized as one possible molecular interaction point during co-infection (Labbé and Saleh, 2008; Lian et al, 2022). This is in line with single-pathogen studies that have highlighted how innate immune signaling is rewired, hijacked or modified by a wide variety of pathogens, which in turn can have downstream effects on secondary pathogens.

## Level 4: molecular mechanisms

With our knowledge of the host–pathogen–pathogen interface continuously expanding, we start getting a glimpse of the diversity of molecular mechanisms that are at play and impact the effect that co-occurring pathogens have. The following chapter will elaborate on the molecular mechanisms that have been described during co-infection and highlight knowledge from single-pathogen infections that we can extrapolate to the co-infection scenario. The insights from single-pathogen studies can be utilized to formulate hypotheses for co-infections, given that pathogens strongly reshape their host and thereby introduce changes to the replicative niche of a secondary pathogen.

## Molecular interaction points for co-infections

As outlined in the introduction, co-occurring pathogens impact each other on various scales, and on each of those, molecular

understanding of the rewiring upon infection is deepening. Most of the knowledge we have is however based on single-pathogen studies which can be extrapolated to the co-infection scenario (Navarro et al, 2005; Jones et al, 2008; Pilar et al, 2012; Sontag et al, 2016; Lian et al, 2022). However, using specific pathogen pairs, and focusing on individual organ systems or host cell types, several interaction points have been investigated (Fig. 3).

These range from alterations to transcription, translation and post-translational modifications (PTMs), which impact secondary pathogens (Walters et al, 2009; LaRock et al, 2015), rewiring of cellular signaling pathways (Srikanth et al, 2011), resource competition and sequestration (Brumell et al, 2002; Philips, 2008), the induction or inhibition of cell death (Matsumoto and Young, 2009; Günster et al, 2017; Maruzuru et al, 2018) to modulation of innate immune responses and immune evasion (Ben-Israel and Kleinberger, 2002; Orzalli et al, 2012; Pilar et al, 2012; Shen et al, 2014). In the following, we discuss these interaction points individually, following the route of infection along its stages.

## Initiation of infection—pathogen colonization, attachment, and invasion

The initiating steps of any infection revolve around colonization—a pathogen reaching a suitable niche and establishing its presence there (Siegel and Weiser, 2015; Pickard et al, 2017)—and subsequent attachment to a permissive host cell. During the first step, pathogens interact with microbiota and encounter extracellular defense mechanisms by the host, such as a mucus layer (Yang et al, 2014). Consequently, overcoming colonization resistance by reshaping the microbiome is a common strategy for pathogens (Gillis et al, 2018; Rogers et al, 2021).

Infections do not occur in an isolated space between a pathogen and its host cell, but rather in the presence of a multitude of other organisms, most predominantly the microbiome, which displays various species across the gut, lung, or skin. Viral pathogens have been shown to interact with the microbiome, hijacking and coating themselves with secreted extracellular vesicles for easier entry (McNamara and Dittmer, 2020; Bello-Morales et al, 2020). It is noteworthy that this form of interaction occurs between the microbes directly, bypassing any involvement of the host.

Subsequent to colonization, attachment to the host cell is essential to initiate infection. In the case of co-occurring pathogens, depending on the specific strain combination, different outcomes have been observed and investigated. In a recent study on IAV co-infections with *Staphylococcus aureus*, the authors showed that the viral infection induces increased presence of the attachment receptor CD47 on the cell surface, thereby promoting attachment of and severity of infection with *S. aureus* (Moon et al, 2024). An even more extreme case of promoting secondary infection is most clearly showcased by the dependence of Hepatitis-D Virus on primary infection with HBV. In the enteric system, Rotavirus has been shown to enhance the invasiveness of facultatively intracellular enterobacteria, such as *Salmonella* and *Shigella* (Bukholm, 1988). For extracellular pathogens, such as *Yersinia*, these early stages of infection are essential to prevent phagocytosis by macrophages, which is mainly driven through injection of the Type-3 secretion system (T3SS) effector YopH (Bliska et al, 1991; Hamid et al, 1999).

Lastly, some studies have investigated the impact of primary bacterial infection on the host cell and how this in turn influences secondary viral infection. In most cases described in literature, this dynamic leads to mutual exclusivity of the two microbes: norovirus is unable to infect macrophages that have been exposed to *Salmonella* (Agnihothram et al, 2015) due to restricted attachment and entry into already bacterially infected cells. *Wolbachia*, while not being pathogenic, excludes infection with Dengue Virus (Fox et al, 2023) through a variety of intracellular pathways that all affect the viral replication, ranging from intracellular transport to innate immune responses (Mushtaq et al, 2024). Further studies are needed to expand our knowledge of the mechanisms underlying this mutual exclusivity, since it provides a promising avenue for the development of novel anti-infectives.

## Niche establishment—intracellular proliferation and intracellular lifestyle

Many pathogens rely on the host cell as a proliferative niche: viruses are dependent on host cell machineries for replication, but also several bacterial or eukaryotic pathogens utilize the host cell to grow, hide from immune detection and spread throughout the host (Douglas, 2009; Silva, 2012; Bahia et al, 2018). To establish this niche and to render it more favorable for proliferation, intracellular pathogens hijack host signaling pathways (Soares-Silva et al, 2016; Watanabe Costa et al, 2016; Haqshenas and Doerig, 2019), translation and transcription machinery (Vijayakumar et al, 2022) and sequester resources, such as nutrients or secondary metabolites from the host (Parrow et al, 2013; Monson et al, 2021). In addition to the impacts this has on the host cell, this rewiring has implications for secondary infections with other intracellular pathogens. Amino acid competition, as well as modification of

lipid trafficking to the replicative niche of *Wolbachia*, albeit not being a pathogen itself, has been linked to the host cell's resistance to Dengue Virus infection (Caragata et al, 2014; Koh et al, 2020).

Viral–viral co-infection plays an important role in recombination and genetic reassortment of respiratory viral infections (Yang et al, 2022; Pipek et al, 2024), as well as Herpes Simplex Virus 1 (HSV-1) (Bowden et al, 2004). Yet also bacterial pathogens impact each other with respect to proliferative behavior: in a recent study, *Chlamydia trachomatis* was shown to display impaired growth and persistence-related behavior during *Neisseria gonorrhoeae* co-infection (Ball et al, 2024).

## Infection detection—immune evasion and modulation of cellular innate immunity

The impact of primary infections on the detection of or innate immune response to a secondary pathogen is heavily understudied. Yet despite this knowledge gap, we have a very good understanding of how single pathogens affect and rewire the host. Despite the scarcity of studies that evaluate the co-infection setting, we can attempt to extrapolate the knowledge of the host–pathogen interface that we have obtained to develop hypotheses how innate immune signaling is relevant for secondary infections. The following paragraphs therefore focus on the mechanistic insights about pathogen sensing and remodeling of the host cell that have been obtained by studying single pathogens. Given the centrality of these key processes, we can assume that they may also play a role during co-infection—the definitive proof for which will have to be provided by individual mechanistic characterization in a co-infection scenario.

Through co-evolution, eukaryotic cells have developed a broad spectrum of mechanisms to detect microbial invaders, each of which is specific to certain pathogen-associated molecular patterns (PAMPs). Toll-like receptors (TLRs) are able to sense bacterial PAMPs, such as lipoproteins (TLR2), lipopolysaccharides (LPS) (TLR4) or flagellin (TLR5), as well as viral PAMPs: dsDNA (TLR3), ssRNA (TLR7 and TLR8), or CpG motifs (TLR9) (Watters et al, 2007). In addition, inflammasomes are activated upon specific inducers, which can be PAMPs: *B. anthracis* lethal factor (NLRP1), T3SS components and flagellin (NAIP/NLRC4), microbial DNA (AIM2), or LPS (caspase-4) (Broz and Dixit, 2016).

Conversely, pathogens have developed molecular strategies to actively modify the host innate immune signaling. HSV-1 directly targets viperin (Shen et al, 2014) or IFI16 (Orzalli et al, 2012) for degradation, and abrogates sensing by cGAS/STING (Su and Zheng, 2017), thereby evading detection in the early infection phase. Other viruses that employ these strategies include Varicella Zoster Virus, causing IRF3 degradation (Zhu et al, 2011) or Pseudorabies Virus, which targets IFNAR1 (Zhang et al, 2017). Additional strategies that viral invaders use to modulate the innate immune response are the alteration of post-translational modifications (PTMs) (Walters et al, 2009; Wang et al, 2013b) as well as sequestration into different intracellular compartments (Yuan et al, 2018; Maruzuru et al, 2018). Interestingly, the insights gained from in vitro studies indeed have implications for the clinical outcome (Dulfer et al, 2023), yet we cannot draw general conclusions on whether viral pre-infection increases or counteracts susceptibility to

secondary infection: while SARS-CoV-2 infection augments the occurrence of subsequent infection with the opportunistic pathogen *Candida albicans* (Tsai et al, 2023), other studies demonstrate that latent HSV-infection can protect from bacterial superinfection due to maintained high levels of IFNγ (Barton et al, 2007).

In addition, various intracellular bacterial pathogens translocate effector proteins into the host cytoplasm that modulate PAMP detection and innate immune responses (Pinaud et al, 2018). Pyroptosis, the inflammatory type of cell death that is most commonly activated during bacterial infection, is dampened or modulated by a variety of T3SS effectors through targeting of inflammasomes by various pathogens: *Shigella* OspC3 targets Caspase-4 and -5 (Kobayashi et al, 2013; Hou et al, 2023), IpgD recodes Calcium signaling, affecting NLRP3 (Sun et al, 2017), and IpaH7.8 acts via glomulin (Suzuki et al, 2014). *Salmonella* and *Shigella* furthermore dampen apoptosis: IpgD or SopB via PI3K/Akt (Cooper et al, 2011; García-Gil et al, 2018; Tran Van Nhieu et al, 2022), VirA via calpain (Bergounioux et al, 2012) or SseK1/3 by interfering in TNFα signaling (Günster et al, 2017). Conversely, *Salmonella* SipB activates apoptosis by targeting Caspases-1 and -2 (Hersh et al, 1999; Jesenberger et al, 2000), thereby highlighting the balance between pro- and anti-death signaling that has strong implications for secondary infections. These model pathogens further translocate proteins that affect pathways that are involved in pathogen detection (IpaH and OspB (Sanada et al, 2012; Fu et al, 2013; Ashida and Sasakawa, 2015)), cell survival (OspF, AvrA, SpvC (Shan et al, 2007; Wei et al, 2012; Jones et al, 2008; Zhu et al, 2007; Yin et al, 2020)) and inflammation (OspB, IpgB2, SopB, SopE/E2 via small GTPases (Klink et al, 2010; Wood et al, 2022; Friebel et al, 2001; Cui et al, 2024)).

Other intracellular pathogens, such as *Legionella* (Luo, 2012), *Burkholderia*, *Listeria* or *Francisella* (Ray et al, 2009) also translocate effector proteins that target crucial signaling pathways that are associated with cell death, such as LegK1 and LnaB activating NF-kB signaling (Ge et al, 2009; Losick et al, 2010) or SdhA dampening cell death to protect *Legionella* in its intracellular niche (Laguna et al, 2006). NF-kB signaling is also one of the predominant pathways targeted by *Salmonella* (Haraga and Miller, 2003; Le Negrate et al, 2008; Pilar et al, 2012; Rolhion et al, 2016; Yang et al, 2021) and *Shigella* (Kim et al, 2005; Ashida et al, 2010; Rahman and McFadden, 2011; Wang et al, 2013a; de Jong et al, 2016; Xian et al, 2024) through a variety of effectors.

Furthermore, attaching extracellular bacteria, such as EPEC, also inject regulators of the innate immune response and cell death through their T3SS (Santos and Finlay, 2015): NleB and NleE have been shown to dampen NF-kB signaling (Nadler et al, 2010; Newton et al, 2010), while EspL blocks necroptosis via RHIM (RIP homotypic interaction motif) proteins (Pearson et al, 2017), and, conversely, EspC has been implicated in enhancing cell death (Serapio-Palacios and Navarro-Garcia, 2016). Similar studies have been performed using *Citrobacter rodentium* in the mouse model (Eng and Pearson, 2021). Furthermore, *Yersinia* targets host cell responses, by YopJ, dampening NF-kB signaling (Ruckdeschel et al, 1998), YopK via NLRP3 (Chen and Brodsky, 2023) or YopM, which reduces activation of the pyrin inflammasome (Ratner et al, 2016). Altering the viability of the host cell, in addition to modulating the actin cytoskeleton via Yop-proteins (Navarro et al, 2005) drastically reshapes the niche that co-occurring pathogens depend on for their proliferation and propagation.

Given this intricate network of detection, defense and rewiring strongly implicates innate immune sensing as a central interaction point for co-occurring pathogens. Since the SARS-CoV-2 pandemic, multiple studies have assessed the impact on innate immunity, and while their number is too large to describe them in detail within the scope of this review, we expect that going forward, SARS-CoV-2 will be a valuable model, for both in vitro and clinical studies, to better understand the impact that co-infections have on single cells, organs and organisms (Maltezou et al, 2023).

# Model systems to study co-infections

To capture the high dimensionality of co-infections and the impact they have on various systems and magnitudes of scale, a broad panel of model systems and study approaches have been developed and employed to investigate co-occurring pathogens. While no model is ideal or can capture the full extent of any given system, there have been various advances to improve their accessibility, usability and versatility to answer the open questions in the field. Broadly speaking, each model system can be placed within a scale ranging from molecular interactions to epidemiological studies, with varying degrees of complexity and physiological relevance (Fig. 4). Since every experimental model system has its own advantages and shortcomings, combining systems across the scales of magnitude, physiological relevance and accessibility is required to gain new insights and connect findings.

## Mathematical models, clinical, and observational studies

To describe the impact of co-infections on ecosystems or entire populations, mathematical models, which have been broadly described to capture single-infection dynamics (Anderson and May, 1979; Siettos and Russo, 2013), are used. Co-infections can be accounted for by changing and adapting the parameters to reflect the co-occurrence of multiple pathogens, and can lead to a more stabilized model, as well as a reduced parameter space for each of the pathogens to dominate (Fenton, 2008). Given that disease outbreaks do not occur in a void space, and that each individual is likely to encounter various pathogens at once, expanding mathematical models to include co-infections is required to better reflect the population-wide outcome and spread. Several modeling approaches have been assessing altered kinetics during Influenza infection, thereby explaining increased susceptibility to secondary bacterial infection (Duvigneau et al, 2016), as well as the impact on co-infections with other viruses (Smith, 2018; Pinky et al, 2023).

These models rely on clinical and population data, which is obtained through observational and clinical studies, the bedrock of co-infections research. Due to the prevalence and the increased severity of co-infections in children, the majority of the literature focuses on this group and spans a variety of infectious agents and endemic areas (Wu et al, 2020; Ibiebele et al, 2023; Babawale and Guerrero-Plata, 2024). Since computational models can only be as good as the data they are trained with, it is imperative that we expand our research in the clinical domain and strive to reduce any inherent biases (certain age groups, (sub-)populations or geographic regions) that exist within the currently available data.

**Figure 4. Model systems used to probe the host–pathogen-pathogen interface.**

On a spectrum that spans clinical studies, cell culture approaches and advanced computational modeling and structure prediction, a vast panel of model systems is used in infection biology. Tailored to the research question at hand, models of high biological complexity with high physiological relevance, as well as simplified or computational approaches that provide accessibility, are available. While no single system is perfect, a Systems Biology approach that combines methodologies on various scales can bridge the gap between physiological context, scalability, accessibility, and clinical relevance. This figure has been adapted from (Walch, 2021).

## In vivo models

Depending on the infectious agent(s) that are being investigated, different in vivo systems are warranted and commonly used. For bacterial infection studies, the murine model remains the most abundant, yet other animal models, such as pig, cow or chicken have been described (Santos et al, 2001; Higginson et al, 2016; Palmer and Slauch, 2017). In addition, invertebrate models, such as *Galleria mellonella* or *Cenorabidis elegans* can serve as easily accessible animal models for precise aspects of the pathogenesis (Higginson et al, 2016; Rong and Liu, 2023).

To study human-specific viral pathogens, one of two strategies are taken for in vivo studies: either a closely related virus is used in an appropriate animal model (e.g., Simian Immunodeficiency Virus in monkeys for HIV-related research), or humanized animals (mostly mice) are infected (Ruiz et al, 2013; Rong and Liu, 2023). While these in vivo models provide a high degree of physiological relevance and are able to reflect the endogenous infection route and dynamics, they lack accessibility and do not allow for the throughput that is needed for systematic or unbiased screening of a multitude of conditions or pairs of co-infecting pathogens.

## Three-dimensional cell culture models and on-a-chip technologies

An alternative to complex in vivo models is presented by novel 3D model systems such as organoids. These stem-cell-derived multi-cellular complexes that resemble the organ of interest can provide an accessible, modifiable, and yet physiological compromise between in vivo and in vitro studies (Aguilar et al, 2021). In the context of co-infection, the most common organoid models are gut or brain (Finkbeiner et al, 2012; Depla et al, 2022) organoids, as well as air-liquid interface models that resemble the lung (Baldassi et al, 2021; Leach et al, 2023; Kurmashev et al, 2024), as well as recent advances into the creation of a lung-on-a-chip (Zamprogno et al, 2021). Novel developments also include apical-out organoid structures that allow easier manipulation to achieve the physiological infection route (Aguirre Garcia et al, 2022; Liebe et al, 2023). Furthermore, on-a-chip models that rely on microfluidics to recapitulate different organs or organ systems have been employed in the study of infections and host–pathogen interactions (Alonso-Roman et al, 2024).

To overcome the technological gap between organoid models, which display high cellular fidelity, yet do not allow for stringent control of the microenvironment, and On-a-Chip systems that are simplified with respect to cell-type representation, but allow for environmental accessibility (Zhao et al, 2024), new tissue engineering advances have explored the concept of organoids-on-a-chip: this has been established for a murine mini-colon (Nikolaev et al, 2020), and very recently for human-derived gastrointestinal models (Mitrofanova et al, 2024). In addition to characterization of the physiological distribution and behavior of the involved cell types, this study also delves into the usability of such systems to probe cytotoxic effects of anti-cancer treatments (Mitrofanova et al, 2024).

Versatile and accessible systems that recapitulate the physiological homeostasis are, going forward, useful models for (co-) infection studies. While there are still improvements required, such as a thorough assessment of the functionality of other cell types encapsulated in the surrounding, such as macrophages, they hold great promise to bridge the gap between the physiological relevance of in vivo work and the accessibility of in vitro cell culture systems. It remains to be seen how implementable these models are to the broader community, or whether their access will remain limited to research groups that hold expertise in tissue engineering.

## In vitro cell culture models and biochemical studies

Therefore, to study cell-type-specific or mechanistic effects, a range of simple, accessible and transferable cell culture models are frequently employed. The most common in vitro systems remain conventional cell culture models, ranging from established cell lines, such as RAW264.7, J774 or THP-1, to epithelial lines, such as Caco-2 cells, human bronchial or nasal epithelial cells, as well as primary cells (bone marrow-derived macrophages, keratinocytes) or immortalized lines that are created therefrom (Verrier et al, 2016; Chanput et al, 2014; López-Jiménez and Mostowy, 2021). The advantage of these cell culture models is the throughput, which allows for multiplexing and the easy assessment of a variety of conditions, as well as the abundance of literature, and hence reference, in which they have been used.

The majority of studies employing these models investigate a mechanistic aspect of the host–pathogen interface, such as the innate immune response, the role of resource limitation or cytoskeletal rearrangements, as discussed in this review. Lastly, to elucidate molecular interactions between the proteins or complexes that are deployed or targeted by pathogens, general biochemical studies, such as protein-protein interactions, as well as continuously improving -omics techniques, such as proteomics (Jean Beltran et al, 2017; Nyman and Matikainen, 2018), transcriptomics (Jorth et al, 2013) or metabolomics (Manchester and Anand, 2017; Bernatchez and McCall, 2020) are employed. Recent developments in this space focus on the resolution, allowing for individual bacteria or single infected host cells to be analyzed and further investigated (Saliba et al, 2016; Suomalainen and Greber, 2021).

Despite their high degree of accessibility and the availability of different systems with a variety of biological characteristics, ex vivo, in cellulo or in vitro studies remain difficult to extrapolate to more complex, multicellular systems. It is therefore necessary to show biological meaningfulness in addition to elucidation of the molecular mechanisms that underlie co-infections. By complementing these various models, we will be able to conduct various forms of co-infection research. Two clearly different strategies exist to expand across biological systems: a bottom-up approach—starting from the molecular mechanism, moving up in complexity to in vivo studies, or a top-down approach—where the starting point is an in vivo or clinical phenotype, which is then dissected in detail by using simplified and more accessible models.

# Conclusion: broadening our understanding of co-infections on various scales is essential

While most molecular and mechanistic studies assess the impact of a single pathogen on the host, this situation occurs rarely in reality. On the contrary, due to prevalent co-occurrence of pathogens, diverse microbiomes and viromes, as well as opportunistic pathogens, infection and host–pathogen interactions are embedded in a complex interplay. Therefore, independently of model systems, pathogens or molecular mechanisms, the co-occurrence of microorganisms and their interactions with a host create a unique complexity and raises many questions (Box 1). Part of that originates from several other dimensions that need to be taken into account when discussing co-infections or the host–pathogen–pathogen interface:

## Directionality

The directionality of interactions has been best described in the field of chemical genomics or when assessing drug-drug

interactions. Generally, interactions can be classified as synergies (a stronger than expected effect), neutral interactions (the baseline assumption of the Bliss Independence model) and antagonisms (a weaker than expected effect). In the case of co-infections, there is no one-size-fits-all solution to the directionality of interactions. As we discussed in this review, there is evidence for synergies and antagonisms, depending on the model system, the pathogens used, and the metric that is quantified.

## (In-)direct effects

The interactions that were described here include those that occur directly between microorganisms, as well as those that involve the host as an intermediary. As we tried to demonstrate, both mechanisms exist, thereby warranting a stringent selection of adequate models, as well as caution in the generalizability of research results.

## Cross-kingdom interactions

The pathogens that were discussed and featured in this work comprise viruses, bacteria, fungi and parasites. Inter-kingdom interactions, and the complexity that arises as a consequence is both daunting and exciting, making it clear that more work has to be done to understand co-infections—be it systematically or mechanistically.

It is, however, essential that we invest more effort into studying this interface, not only to better understand clinical observations and outcomes, and to pave the way for novel anti-infective treatment strategies for co-infections (Box 1). The knowledge we generate from co-infections also has implications on infectious disease research in general: microbes are excellent cellular biologists, and after thousands or millions of years of co-evolution with their hosts, have developed intricate strategies to rewire how cells function, signal, perceive their environment and even die. Harnessing this, to develop new strategies to combat infections, as well as pathophysiologies that are host-specific, such as cancer, chronic inflammation or immunodeficiency, is a potent way forward and remains an essential research endeavor (Box 1).

## Peer review information

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

## Acknowledgements

The authors would like to thank the Inkscape project for providing easily usable, license-free vector graphic editing software, which was used to generate the sketches for the figures shown in this review. The authors furthermore acknowledge Paperpile for generating and managing references throughout the text. Funding for this work has been provided by the Swiss National Science Foundation for both PW (TMPFP3_217085) and PB (310030_192523).

## Author contributions

**Philipp Darius Konstantin Walch**: Conceptualization; Funding acquisition; Visualization; Writing—original draft; Writing—review and editing. **Petr Broz**: Conceptualization; Funding acquisition; Project administration; Writing—review and editing.

## Disclosure and competing interests statement

The authors declare no competing interests.

