## [Peer Review File · EMBO Reports]

Molecular Mechanisms of Co-Infections

Philipp Walch and Petr Broz

Corresponding author(s): Petr Broz (petr.broz@unil.ch) , Philipp Walch (Philipp.Walch@unil.ch)

Review Timeline:

Submission Date:	3rd Mar 25
Editorial Decision:	28th Mar 25
Revision Received:	29th May 25
Accepted:	24th Jun 25

Editor: Achim Breiling

Transaction Report:

Dear Petr, Dear Philipp,

Thank you for the submission of your review article to our editorial offices. I have now received the full set of referee reports that is copied below. As you will see, all three referees state that your manuscript is interesting and timely. They have several suggestions to improve the submission that I kindly ask you to address in a final revised manuscript.

Given the constructive referee comments, I would thus like to invite you to revise your manuscript with the understanding that all referee points will be addressed in the revised manuscript and in a detailed point-by-point response.

I further have these editorial requests:

- Please add author affiliations and indicate corresponding authors, including e-mail addresses below the title.
- Please provide a final abstract with not more than 175 words.
- Please add up to 5 keywords to the manuscript and place these below the abstract.
- We have space for 1 more figure, and it would be nice to have indeed 4 figures, as we encourage authors to maximize the use of visual elements, which will increase the accessibility of the piece to a non-specialist readership. Please consider adding 1 more figure and note the instructions regarding figures below.
- We updated our journal's competing interests policy in January 2022 and request authors to consider both actual and perceived competing interests. Please review the policy <https://www.embopress.org/competing-interests> and update your competing interests if necessary. Please name this section 'Disclosure and Competing Interests Statement' and put it after the discussion, before the references.
- Do you want to add Acknowledgements? In case, please provide this section before the 'Disclosure and Competing Interests Statement'. Please make sure that any funding information provided in the Acknowledgements is also entered into the online submission system and that it is complete and similar to the one in the acknowledgement section of the manuscript text file.
- Please also note our reference format:
<http://www.embopress.org/page/journal/14693178/authorguide#referencesformat>
- We usually ask our authors to include a box called "In need of answers" that briefly outlines the major questions that are still open in a given field in the form of a few bullet points. These questions can be accompanied by a brief explanation of what would be needed to address them and may provide helpful towards setting the stage for future experimentation in the field. For an example see this recent review we published: <https://www.embopress.org/doi/full/10.1038/s44319-024-00135-4>
- Please also add callouts for the box to the manuscript text (Box 1).
- Please move the table to the end of the manuscript text file, before Box 1.

I think this is a very interesting review and while I appreciate that incorporating the referees' suggestions will still require some work, I am convinced that the article is worth it and will benefit from it.

When submitting your revised manuscript, we will require a Microsoft Word file (.doc) of the revised manuscript text including detailed figure legends (placed after the references), tables, but without the figures.

Please provide the final figures as separate, high-resolution files (without their legends) as .pdf, .eps, .tif, or .jpg (one file per figure). Please finalize the drafts provided and make sure they accurately illustrate the key scientific concepts that you wish to show.

Please also note the following points:

- If there are certain aspects of your figure draft that are based upon assumptions or where the scientific data remains ambiguous (for example, schematically depicting a presumed direct protein-protein interaction, protein shape or subcellular localizations etc.) please add a comment so that we can work with you on an accurate depiction. Please ensure the directionality and nature of interactions is presented accurately.

- If the figure or single panels of the figure have been adapted from a published figure, please add this information to the figure legend (e.g., 'Adapted from...' or 'Based on...'). The editor will discuss if a reference and permission will be necessary

- Please only re-use figures or parts of a figure if this is essential for understanding the concept communicated. Often a reference to a previous paper will suffice. If the figure contains re-used images or elements of images, including schematics, micrographs or photos, please make sure that you have the permission/license to publish it (this also applies to your own previous work, if the journal you published in retains copyright. Certain 'creative commons' open access licenses, such as CC-BY 4.0, allow re-use without additional formal permissions). All re-used material must be explicitly cited.

- If you use an image data base for scientific iconography (e.g., BioRender), please let us know if you have a license that allows for publication in an academic journal. Often authors use misleading iconography for expedience. Please ensure the information shown is scientifically accurate. If in doubt, please discuss with the editor or provide a sketch so that our designers can create accurate iconography. Please acknowledge the use of BioRender once in the Acknowledgements section (not in the figure legend).

- For figures created using a software for editing vector objects like Inkscape, CorelDraw etc., please send the file as a PDF (or SVG, or EPS), PowerPoint or Keynote in which the labels and objects are still editable. For figures created using Adobe Illustrator, please send the Illustrator (.ai) file.

I look forward to seeing a revised version of your manuscript when it is ready. Please let me know if you have questions or comments regarding the revision.

Please use this link to submit your revision: <https://embor.msubmit.net/cgi-bin/main.plex>

Kind regards,

Achim

Referee #1:

Co-infections with multiple pathogens are clinically highly relevant, however the underlying molecular pathology is hardly understood. Thus, a review about this topic is important, timely and relevant. Here, the authors summarise what is currently known about the epidemiological and molecular interactions and the clinical impact of co-infection. The idea to define different layers is creative and useful. I also really appreciate that the authors also discuss model systems that could enable more insights into host-pathogen-pathogen interaction scenarios. Overall, it is a well written review and highlights the dire need for future research on the (molecular) and implications of co-infections.

Please find below a few suggestions to improve the manuscript:

- I wonder whether more information could be included on what is known on the clinical outcome of bacterial/viral or viral/viral coinfections. There is a body of literature available, which could also help to provide information for a table summarizing positive or negative interference and clinical implications

- Occasionally the writing could be improved, or rather, more streamlined and focused. Sentences like "The scales of magnitude that this review embarks on in the following ... implications on several or even all levels." "The following chapter will ...that we can extrapolate to the co-infection scenario." "As outlined in the prior chapter... upon infection is deepening" could be omitted/shortened. Some sentiments, despite being important, are repeated multiple/too many times "yet we are only beginning to investigate ... cell-to-cell communication." "The insights from single-pathogen studies ... introduce changes to the replicative niche of a secondary pathogen." In the abstract lines 1-4 could be streamlined, the wording making it sound unnecessarily convoluted. Redundancies should be eliminated. CD47 is referenced/explained at least twice. Similarly the interaction/dependence of HepD on HepB.

- In the introduction I would suggest to turn the explanation around, starting from the larger more accessible interactions (epidemiology) to the more complex molecular interactions.

- A categorization of the clinical outcome should be made/stated early in the manuscript. The abstract only mentions exacerbation, but more outcomes can be envisioned. It should also be addressed as separate point (maybe with the help of a table, see above), as it does not fit too well into the 'level' categorization.

- The prominent examples chapter could also be separated from the introductory sentences (and not under level 1).

- It would be good to directly name the examples listed in "For virus-virus coinfections, mutual exclusivity has been described ... or synergies (Abdullah et al. 2017)."

- Viruses could be mentioned in the niche establishment as well.

- "the combined effect of a co-infection is even more deleterious" is this always the case?
- The molecular mechanisms chapter seems to span from epidemiological interactions to cell-cell to within one cell. E.g. the establishment of a niche within the microbiome. Thus it is not fitting that well into the current level outline.
- Am not entirely sure whether the detailed table on bacterial effector functions is necessary to get the point across. Alternatively, the paragraph describing the putative interactions has to be expanded to really make use of the information contained in the table and highlight this interaction as a well studied example.
- Figure 2 - might benefit from colour coded or shaded areas and arrows where the interactions could be easily identified. For the 2nd and 3rd interactions its difficult to locate an epicentre or upstream signal
- Although mentioned in Figure 3, in silico modelling on a molecular scale is also not really discussed in the text. Alternatively, Figure 3 could be simplified according to the broader categories in the text (which I would prefer).
- Epidemiology paragraph: Was it ever considered in published models as a perturbation of infection dynamics?
- Maybe as a suggestion for the conclusions: Single pathogen infections are just a reduced representation of reality (easy, but maybe misleading model). Normally, co-infections are physiological (we all have a microbiome, we have Herpesviruses, usually several of them). Current techniques as well as models may allow studying these multi-infections approaching the real biology a little better.

Referee #2:

In their review „Molecular Mechanisms of Co-Infections“, the authors address a highly relevant topic at the interface of microbiology, immunology and epidemiology. The scope of this review extends beyond most articles focusing on specific pathogens, organ systems or scales of magnitude such as epidemiology or specified molecular pathways. This is a plus that might not be reflected adequately in the title of the article, that suggests a mere focus on molecular mechanisms. Please find below further comments, addressing of which might strengthen the article.

- The first two paragraphs are fully lacking references. Referencing suitable articles would strengthen the points made, e.g. co-evolution of pathogens and hosts.
- In figure 1 it could be misleading that only virus is transferred between individuals (level 1). For level 3 it is not really clear, which processes are depicted. This could be improved via labeling in the figure or a more detailed figure legend.
- *Clostridium difficile* should be called *Clostridioides difficile*
- The authors state that "Interestingly, in the case of SARS-CoV-2, these impacts can outlast the primary viral infection beyond its clearance". This is also the case for other respiratory viruses such as influenza A virus, as described e.g. in doi: 10.1084/jem.20070891
- In the Level 2 paragraph about epithelial destruction in influenza infection predisposing for bacterial infection also immune mediated mechanisms should be mentioned such as the reference given in the above comment or DOI: 10.1128/IAI.00298-15
- In the context of "The latter has been prominently described for the BCG-vaccine, which, in addition to conferring protection to tuberculosis, also modifies the responses of innate immune cells, leading to an increased protection towards a variety of secondary infections" the term trained immunity could be provided and defined.
- With respect to "pathogens actively modulate the extracellular niche of colonized organs and tissues", this could be extended again to influenza A virus: doi: 10.3389/fcimb.2021.643326.
- In Figure 2 authors should explain PTM in the legend (even though it is written later in the text).
- The "Molecular interaction points for co-infections" section and also table 1, at times seems distracted from the focus of co-infections-The authors provide many basic mechanisms and pathways and it does not always become clear, if and how these have implications for co-infections. If authors could state and provide references for such implications, this would support the relevance of the discussed pathways/mechanisms with respect to the topic of the manuscript. The same holds true for the paragraph describing model systems, that could be more focused on their use in co-infection studies.
- In figure 3 the text outside the box seems very small.
- With respect to mathematical models, the authors could also refer to mathematical modelling approaches from experimental results in co-infection, such as e.g. doi: 10.1111/imr.12692, DOI: 10.3390/v15061303, DOI: 10.1038/srep37045

Referee #3:

A very comprehensive review and will be of immense interest conceptually but supported by a rich diversity of example literature, as well as bringing a forward-looking perspective. I have only few comments/suggestions:

- Is there a need to define a co-infection versus a super-infection as super-infection are more widely known, especially in flu literature.
- I see little mention about protective viral-viral interactions, such as rhinovirus and SARS2 (Human Rhinovirus Infection Blocks Severe Acute Respiratory Syndrome Coronavirus 2 Replication Within the Respiratory Epithelium: Implications for COVID-19 Epidemiology | The Journal of Infectious Diseases | Oxford Academic
- For HIV bits - please state in untreated cases
- Why focus of table 1 on bacterial effectors and not viral? This review is in parts comprehensive and broad and in others focused on bacterial>viral

- Is malaria discussed here? Should be common and important co-infection
- Seems human focused but lots of literature on livestock infections outwith their role as model systems - why not use analogous models for example, swine influenza in pigs and bacterial pathogens etc
- Check the consistency of capitalisation of viral names
- Explain: As outlined in the prior chapter -= what prior chapter?

Editorial requests:

- Please add author affiliations and indicate corresponding authors, including e-mail addresses below the title.

We added the required information.

- Please provide a final abstract with not more than 175 words.

We shortened the abstract to 175 words.

- Please add up to 5 keywords to the manuscript and place these below the abstract.

We added keywords that we find relevant to the review.

- We have space for 1 more figure, and it would be nice to have indeed 4 figures, as we encourage authors to maximize the use of visual elements, which will increase the accessibility of the piece to a non-specialist readership. Please consider adding 1 more figure and note the instructions regarding figures below.

We added an additional figure on common clinical co-infection pairs, and adjusted the referencing accordingly.

- We updated our journal's competing interests policy in January 2022 and request authors to consider both actual and perceived competing interests. Please review the policy <https://www.embopress.org/competing-interests> and update your competing interests if necessary. Please name this section 'Disclosure and Competing Interests Statement' and put it after the discussion, before the references.

We added the respective section

- Do you want to add Acknowledgements? In case, please provide this section before the 'Disclosure and Competing Interests Statement'. Please make sure that any funding information provided in the Acknowledgements is also entered into the online submission system and that it is complete and similar to the one in the acknowledgement section of the manuscript text file.

We added the respective section and also acknowledged funding sources.

- Please also note our reference format:
<http://www.embopress.org/page/journal/14693178/authorguide#referencesformat>

We changed the referencing format to "EMBO reports" in Paperpile, which we use for generating references. We hope that this is now accurate.

- We usually ask our authors to include a box called "In need of answers" that briefly outlines the major questions that are still open in a given field in the form of a few

bullet points. These questions can be accompanied by a brief explanation of what would be needed to address them and may provide helpful towards setting the stage for future experimentation in the field. For an example see this recent review we published: <https://www.embopress.org/doi/full/10.1038/s44319-024-00135-4>

- Please also add callouts for the box to the manuscript text (Box 1).

We added a “in need of answers” box and included callouts in the text.

- Please move the table to the end of the manuscript text file, before Box 1.

We removed the table in the revised manuscript and replaced it with a text paragraph (upon suggestion of the reviewers).

- When submitting your revised manuscript, we will require a Microsoft Word file (.doc) of the revised manuscript text including detailed figure legends (placed after the references), tables, but without the figures.
- Please provide the final figures as separate, high-resolution files (without their legends) as .pdf, .eps, .tif, or .jpg (one file per figure). Please finalize the drafts provided and make sure they accurately illustrate the key scientific concepts that you wish to show.

We will adhere to this formatting and will submit the files accordingly.

Please also note the following points:

- If there are certain aspects of your figure draft that are based upon assumptions or where the scientific data remains ambiguous (for example, schematically depicting a presumed direct protein-protein interaction, protein shape or subcellular localizations etc.) please add a comment so that we can work with you on an accurate depiction. Please ensure the directionality and nature of interactions is presented accurately.
- If the figure or single panels of the figure have been adapted from a published figure, please add this information to the figure legend (e.g., 'Adapted from...' or 'Based on...'). The editor will discuss if a reference and permission will be necessary
- Please only re-use figures or parts of a figure if this is essential for understanding the concept communicated. Often a reference to a previous paper will suffice. If the figure contains re-used images or elements of images, including schematics, micrographs or photos, please make sure that you have the permission/license to publish it (this also applies to your own previous work, if the journal you published in retains copyright. Certain 'creative commons' open access licenses, such as CC-BY 4.0, allow re-use without additional formal permissions). All re-used material must be explicitly cited.
- If you use an image data base for scientific iconography (e.g., BioRender), please let us know if you have a license that allows for publication in an academic journal. Often

authors use misleading iconography for expedience. Please ensure the information shown is scientifically accurate. If in doubt, please discuss with the editor or provide a sketch so that our designers can create accurate iconography. Please acknowledge the use of BioRender once in the Acknowledgements section (not in the figure legend).

- For figures created using a software for editing vector objects like Inkscape, CorelDraw etc., please send the file as a PDF (or SVG, or EPS), PowerPoint or Keynote in which the labels and objects are still editable. For figures created using Adobe Illustrator, please send the Illustrator (.ai) file.

We are confident that we adhere to these points. The figures were generated in Inkscape, and .svg files will be provided. We furthermore did not use any licensed material, but created all shapes from hand or from license-free vector graphic databases.

Referee #1:

Co-infections with multiple pathogens are clinically highly relevant, however the underlying molecular pathology is hardly understood. Thus, a review about this topic is important, timely and relevant. Here, the authors summarise what is currently known about the epidemiological and molecular interactions and the clinical impact of co-infection. The idea to define different layers is creative and useful. I also really appreciate that the authors also discuss model systems that could enable more insights into host-pathogen-pathogen interaction scenarios. Overall, it is a well written review and highlights the dire need for future research on the (molecular) and implications of co-infections. Please find below a few suggestions to improve the manuscript:

I wonder whether more information could be included on what is known on the clinical outcome of bacterial/viral or viral/viral coinfections. There is a body of literature available, which could also help to provide information for a table summarizing positive or negative interference and clinical implications

We appreciate the comment and agree that there is a large number of clinical studies available in the literature. Given the focus of the review on molecular mechanisms, we decided to only briefly discuss the clinical implications, however, we included an additional figure (Figure 2) which summarizes the most commonly detected co-infections, and we have added minor edits to the text that address the clinical outcome.

Occasionally the writing could be improved, or rather, more streamlined and focused. Sentences like "The scales of magnitude that this review embarks on in the following ... implications on several or even all levels." "The following chapter will ...that we can extrapolate to the co-infection scenario." "As outlined in the prior chapter... upon infection is deepening" could be omitted/shortened. Some sentiments, despite being important, are repeated multiple/too many times "yet we are only beginning to investigate ... cell-to-cell communication." "The insights from single-pathogen studies ... introduce changes to the replicative niche of a secondary pathogen." In the abstract lines 1-4 could be streamlined, the wording making it sound unnecessarily convoluted. Redundancies should be eliminated. CD47 is referenced/explained at least twice. Similarly the interaction/dependence of HepD on HepB.

We apologize for the imperfect style and have tried to ameliorate and streamline our formulation in the manuscript. We further tried to remove redundancies, however left some in on purpose, since they serve as a guiding theme throughout the manuscript to discuss the different levels that are impacted by co-infections.

In the introduction I would suggest to turn the explanation around, starting from the larger more accessible interactions (epidemiology) to the more complex molecular interactions.

We rearranged the first part of the introduction, but maintained the clinical outcome as starting point, since it is the motivation to better understand the molecular mechanisms.

A categorization of the clinical outcome should be made/stated early in the manuscript. The abstract only mentions exacerbation, but more outcomes can be envisioned. It should also be addressed as separate point (maybe with the help of a table, see above), as it does not fit too well into the 'level' categorization.

We agree that the clinical outcome that is mentioned focuses on the exacerbation of symptoms. We have added more qualifying statements, including the acknowledgement that there is a bias towards exacerbated outcomes, since a symptom amelioration during co-infection may remain hidden due to lack of hospitalization. However, we would again like to stress that we want to focus on the molecular mechanisms, rather than the clinical outcome during co-infections.

The prominent examples chapter could also be separated from the introductory sentences (and not under level 1).

We hope that the newly added Figure 2 responds to this comment.

It would be good to directly name the examples listed in "For virus-virus coinfections, mutual exclusivity has been described ... or synergies (Abdullah et al. 2017)."

We have added the respective pathogens to each of the studies and examples.

Viruses could be mentioned in the niche establishment as well.

We added some information about viruses to this section

"the combined effect of a co-infection is even more deleterious" is this always the case?

We apologize for phrasing this too broadly, and have qualified the generalizability of this statement.

The molecular mechanisms chapter seems to span from epidemiological interactions to cell-cell to within one cell. E.g. the establishment of a niche within the microbiome. Thus it is not fitting that well into the current level outline.

In this chapter, we seek to highlight molecular mechanisms that have been described in the context of coinfection. While they go beyond the single-cell level, we nonetheless found those that occur on the organ-specific level (e.g. colonization and competition with the microbiome) to be of interest to the reader. In this section, we do however not focus on epidemiological effects of co-infections and therefore humbly disagree with the statement that this section does not fit its title.

Am not entirely sure whether the detailed table on bacterial effector functions is necessary to get the point across. Alternatively, the paragraph describing the putative

interactions has to be expanded to really make use of the information contained in the table and highlight this interaction as a well studied example.

We agree with this assessment and therefore replaced the table with a short paragraph summarizing the findings. We furthermore indicate that the knowledge described by these studies stems from single-pathogen infections and therefore only has indirect implications for the co-infection case.

Figure 2 - might benefit from colour coded or shaded areas and arrows where the interactions could be easily identified. For the 2nd and 3rd interactions its difficult to locate an epicentre or upstream signal.

Apologies for not making this figure more visually clear. We have now added numbers to the different sections and furthermore expanded the figure legend to provide additional clarity.

Although mentioned in Figure 3, *in silico* modelling on a molecular scale is also not really discussed in the text. Alternatively, Figure 3 could be simplified according to the broader categories in the text (which I would prefer).

We changed figure 3, and amended the *in silico* part to include disease modeling. We furthermore added some additional points and references to the respective text section. We hope that thereby, the text and figure are cohesive and informative.

Epidemiology paragraph: Was it ever considered in published models as a perturbation of infection dynamics?

To our knowledge, there are studies that have looked at the perturbation that co-infections have on disease kinetics and infection dynamics. We included several of these studies in the revised manuscript.

Maybe as a suggestion for the conclusions: Single pathogen infections are just a reduced representation of reality (easy, but maybe misleading model). Normally, co-infections are physiological (we all have a microbiome, we have Herpesviruses, usually several of them). Current techniques as well as models may allow studying these multi-infections approaching the real biology a little better.

Thank you for that suggestion. We attempted at underlining the physiological relevance of co-infections throughout the manuscript, including in the conclusion in this revised version.

Referee #2:

In their review „Molecular Mechanisms of Co-Infections“, the authors address a highly relevant topic at the interface of microbiology, immunology and epidemiology. The scope of this review extends beyond most articles focusing on specific pathogens, organ systems or scales of magnitude such as epidemiology or specified molecular pathways. This is a plus that might not be reflected adequately in the title of the article, that suggests a mere focus on molecular mechanisms. Please find below further comments, addressing of which might strengthen the article:

The first two paragraphs are fully lacking references. Referencing suitable articles would strengthen the points made, e.g. co-evolution of pathogens and hosts.

We have added references to these sections, but refrain from citing studies for every claim, since these paragraphs mainly summarize points that we raise throughout the manuscript, where we cite studies accordingly.

In figure 1 it could be misleading that only virus is transferred between individuals (level 1). For level 3 it is not really clear, which processes are depicted. This could be improved via labeling in the figure or a more detailed figure legend.

We apologize for the confusion and have changed the figure according to your suggestion. We have further amended the figure legend to include more information.

Clostridium difficile should be called *Clostridioides difficile*.

Thank you! We changed the species name accordingly.

The authors state that "Interestingly, in the case of SARSCoV-2, these impacts can outlast the primary viral infection beyond its clearance". This is also the case for other respiratory viruses such as influenza A virus, as described e.g. in doi: 10.1084/jem.20070891

Thank you for the suggestion, we added the reference to this interesting work.

In the Level 2 paragraph about epithelial destruction in influenza infection predisposing for bacterial infection also immune mediated mechanisms should be mentioned such as the reference given in the above comment or DOI: 10.1128/IAI.00298-15

We added a sentence that acknowledges and references this.

In the context of "The latter has been prominently described for the BCG-vaccine, which, in addition to conferring protection to tuberculosis, also modifies the responses of innate immune cells, leading to an increased protection towards a variety of secondary infections" the term trained immunity could be provided and defined.

We apologize for not having acknowledged and define this term – we have now added this to the revised manuscript.

With respect to "pathogens actively modulate the extracellular niche of colonized organs and tissues", this could be extended again to influenza A virus: doi: 10.3389/fcimb.2021.643326.

We thank the reviewer for this suggestion and added it to the manuscript.

In Figure 2 authors should explain PTM in the legend (even though it is written later in the text).

We changed the figure, and removed all abbreviations.

The "Molecular interaction points for co-infections" section and also table 1, at times seems distracted from the focus of co-infections-The authors provide many basic mechanisms and pathways and it does not always become clear, if and how these have implications for co-infections. If authors could state and provide references for such implications, this would support the relevance of the discussed pathways/mechanisms with respect to the topic of the manuscript. The same holds true for the paragraph describing model systems, that could be more focused on their use in co-infection studies.

We agree with the assessment and therefore replaced Table 1 with a short paragraph summarizing the key molecular interaction points. Unfortunately, there are only few studies on the impact of innate immune signaling during co-infection, which is why we decided to consult single pathogen studies that highlight perturbations of the host cell, since these have a downstream effect on secondary pathogens. We further qualify that these hypotheses don't necessarily have direct evidence from co-infection studies, but nonetheless hold relevance for a better understanding of the pathogen-pathogen interplay.

In figure 3 the text outside the box seems very small.

We changed the figure, so that it is more readable.

With respect to mathematical models, the authors could also refer to mathematical modelling approaches from experimental results in co-infection, such as e.g. doi: 10.1111/imr.12692, DOI: 10.3390/v15061303, DOI: 10.1038/srep37045

Thank you for suggesting these studies – we amended the manuscript accordingly.

Referee #3:

A very comprehensive review and will be of immense interest conceptually but supported by a rich diversity of example literature, as well as bringing a forward-looking perspective. I have only few comments/suggestions:

Is there a need to define a co-infection versus a super-infection as super-infection are more widely known, especially in flu literature.

We added a sentence explaining the difference in the introduction section. However, for this review, we believe that both scenarios, co-infection and superinfection are of very high interest.

I see little mention about protective viral-viral interactions, such as rhinovirus and SARS2 (Human Rhinovirus Infection Blocks Severe Acute Respiratory Syndrome Coronavirus 2 Replication Within the Respiratory Epithelium: Implications for COVID-19 Epidemiology | The Journal of Infectious Diseases | Oxford Academic

In the revised manuscript, we now cite the study that you suggested, alongside other examples of viral-viral mutual exclusivity, which have often been investigated using plant models.

For HIV bits - please state in untreated cases

We changed the text accordingly.

Why focus of table 1 on bacterial effectors and not viral? This review is in parts comprehensive and broad and in others focused on bacterial>viral

The revised manuscript has been changed to not contain Table 1, but rather summarize the information in a brief paragraph. We hope that in doing so, we balance the focus and also give less space to single-pathogen infection models.

Is malaria discussed here? Should be common and important co-infection

Indeed, malaria is a highly relevant infectious agent during co-infections. We had briefly mentioned it in conjunction with HIV, but have now expanded this section to include other malaria co-infections.

Seems human focused but lots of literature on livestock infections outwith their role as model systems - why not use analogous models for example, swine influenza in pigs and bacterial pathogens etc

We agree that this review focuses on human pathogens – mainly for reasons of limited space. It is true that co-infections play a big role in other systems as well, and we mention plant models, as well as livestock models at different sections of the review.

Check the consistency of capitalisation of viral names

Thank you, we now capitalize all viral names.

Explain: As outlined in the prior chapter -= what prior chapter?

We changed the phrasing to specify "introduction".

Prof. Petr Broz
University of Lausanne
Immunobiology
Chemin des Boveresses
Epalinges 1066
Switzerland

Dear Prof. Broz,

I am pleased to inform you that your review manuscript has been accepted for publication in EMBO reports. Your manuscript will be processed for publication by EMBO Press. It will be copy edited and you will receive page proofs prior to publication.

You will soon be contacted by Springer Nature to sign your publishing license. When you login to the customer service website, please use the token/code copied below to waive the article publication charges. Should you experience any difficulty, please email publishing@embo.org.

LTE3NTQ0MDM3MJG

If you have any further questions, please do not hesitate to contact the Editorial Office. Thank you for your contribution to EMBO Reports.

Yours sincerely,
